# Achieving Carbon Neutrality through Urban Planning and Design

**DOI:** 10.3390/ijerph20032420

**Published:** 2023-01-29

**Authors:** Zhiqiang Wu, Zichen Zhao, Wei Gan, Shiqi Zhou, Wen Dong, Mo Wang

**Affiliations:** 1College of Architecture and Urban Planning, Tongji University, Shanghai 200092, China; 2College of Design and Innovation, Tongji University, Shanghai 200093, China; 3Shanghai Tongji Urban Planning & Design Institute Co., Ltd., Shanghai 200092, China; 4College of Architecture and Urban Planning, Guangzhou University, Guangzhou 510006, China

**Keywords:** urban building cluster, carbon emission, carbon sink, carbon neutrality, intervention measures

## Abstract

Much of the research on climate change has focused on carbon reduction in cities or countries. However, more attention needs to be paid to how to achieve carbon neutrality in the urban design and planning stage, and the lack of quantitative analysis of carbon related to urban space makes it difficult to locate urban space and provide direct guidance for urban planning and design. This study proposed three optimization paths to achieve carbon neutrality in multi-scale urban building clusters. Firstly, we reconstructed the quantitative calculation system of urban building communities with the goal of carbon neutrality; secondly, we screened the carbon source reduction and carbon sink interventions that are suitable for multi-scale urban building communities; finally, we constructed a carbon emission and carbon sink calculation system of planning and design schemes based on the layout of relevant elements of planning and design schemes with a grid cell of 100 × 100 m. In practice, there was a gap of about 115,000 tons of CO_2_ from the carbon-neutral target and 26% of carbon emission was distributed in the Xiajiabian Station TOD. In this study, nine types of carbon reduction measures were adopted to achieve carbon neutrality in the region, among which the highest carbon reduction was achieved by biomass energy measures, accounting for 29% of the total carbon reduction of 33,745.27 T. The objective of this study is to accurately and quantitatively assess the carbon targets of urban spaces at different scales and adopt effective measures to achieve carbon neutrality.

## 1. Introduction

Carbon neutrality is achieved by applying carbon reduction related technologies to offset greenhouse gases generated by human activities within a specific period, and the built environment of cities is a major source of greenhouse gases. Therefore, urban planning and design with a carbon-neutral perspective, as well as the layout of effective carbon reduction schemes in the urban design phase, are crucial to sustainable urban development [1]. In addition, building clusters at multiple scales play an essential role in carbon reduction pathways as they constitute the central urban units. Moreover, quantitative evaluation is one of the most effective pathways towards achieving carbon neutrality goals [2]. Therefore, an accurate multi-scale quantitative study of carbon emissions from building clusters can facilitate the development of urban carbon neutrality implementation pathways [3].

During the urban planning process, urban planners and governments set policy and strategic goals, and these decisions and plans significantly affect the energy efficiency and carbon emissions of cities. However, there need to be more quantitative indicators and intervention strategies for locatable spaces to assist urban planners and policymakers in achieving urban low-carbon development goals at the planning and design stages. In the existing studies on urban CO_2_ estimation, some scholars use empirical CO_2_ equations and carbon assessment software to measure the values and impacts of urban CO_2_. Others precisely assess the current CO_2_ impact factors with neighborhoods or streets as the study unit. Moreover, some studies include carbon sink or source reduction in the study of carbon reduction measures.

The empirical formula of carbon emission is constructed by studying the influence of urban elements on carbon emission. Urban features mainly include buildings, transportation and citizens. Empirical formulas are usually divided into two types: carbon emission calculations and carbon reduction calculations (Table 1). For example, ZhangJie (2015) uses direct CO_2_ emissions from urban residents in China as urban carbon emissions. Chao Liu (2017) constructs a model to calculate carbon emissions (buildings, transportation) and carbon sinks to explore the relevance of integrated urban carbon dioxide (CO_2_) emission models and CO_2_ mitigation plans at the new urban scale [4]. Congrong (2018) uses urban land-related carbon emissions sum as urban carbon emissions [5]. S. Zubelzu (2015) explored the projected urban emissions using a household-based carbon footprint correlation study with urban design parameters (area, family unit) [6]. Both of the above carbon assessment methods are urban-scale carbon estimation and do not consider small-scale assessment of urban building clusters. In recent years, some studies have started to try carbon emission estimation of local urban space, mainly divided into urban single-site space scale, street space and internal or local space of buildings, for example, Cong R. (2018) pointed out that carbon emission estimation of different site spaces, such as commercial land use leads to urban carbon emission increase [5]. Street-scale carbon emissions have been extensively studied. For example, J Felkner et al. (2021) studied the effect of the scale of building clusters within streets on urban energy efficiency and carbon emissions [7]. Vititneva (2021) investigated the impact of street space composition on the amount of carbon dioxide emissions [8]. Baghi (2021) investigated the elements of carbon emission correlation at the city block scale [9]. MariHukkalainen (2017) explored the relationship between urban building clusters and transportation and carbon emission calculation based on the Kurke platform aid [10]. Stojanovski T. (2019) proposed and tested a model of urban travel space selection, thus impacted on urban carbon emission [11]. Zhang Xiaoping (2021), Hasan, Javeriya (2021, Christina Meier-Dotzler1 (2021) and Wang Yifan et al. (2021) empirically tested to explore the energy and emission impacts of building spatial morphology and the full-cycle operation of internal space [12,13,14,15]. Hasanan and Javeriya (2021) used a correlation study between roof space and solar potential to explore how roof space changes building carbon emissions [14]. However, the above studies still have limitations for urban carbon dioxide measurement. First, most of the above studies analyze single variables of urban building clusters, such as roads or buildings, and prove their impact on urban carbon dioxide emissions. Still, they do not include multiple variables of urban building clusters in the carbon dioxide assessment mechanism. Second, although the calculation of carbon emissions began to focus on micro-scale studies, most studies concentrated only on individual buildings and did not include the linkages between building groups when assessing impact factors.

Research on urban carbon reduction calculation is mainly divided into two calculation approaches: urban carbon sink and urban carbon source carbon reduction. However, most urban carbon reduction approaches remain at the macroscopic research or policy development stage. For example, HuanyuZheng (2020) pointed out that urban carbon source carbon reduction mainly refers to the curbing effect of renewable energy on carbon emission intensity. This study also verified that for every 1% of renewable energy development, carbon emission intensity decreases by 0.028–0.043% [16]. However, most of the studies focus on urban carbon reduction policy development, such as MariHukkalainen (2017), who proposed carbon reduction measures such as building and transportation related as well as coordinated targets for each governmental component [10]. Research on urban carbon sink measures has mainly focused on the influence of green space types on carbon sink values. For example, Huanyu Zheng (2021) proposed that urban carbon sinks mostly refer to natural carbon sinks. Natural carbon sink refers to the absorption and transformation of carbon by green space systems such as cropland, woodland, garden, grassland, wetland, green roof and three-dimensional greenery [17,18]. However, the study of urban carbon sink measures is a unique study of plants and green space types, which is mainly conducted by actual local measurements and lacks sure accuracy and ease of implementation [17,19,20,21,22].

**Table 1 ijerph-20-02420-t001:** Summary of urban carbon emission calculation methods.

Author, Research Year	Carbon Assessment Objects	Carbon Assessment Formula	Assessing the City Factors
Zhang Jie, 2015 [23]	Direct CO_2_ emissions from urban residents in China	$EDCE=E_{1}+EF_{1i}$ $GDCE=\overset{r}{\underset{i=1}{\sum }}F_{r}\times NV_{r}\times EF_{r}$ $TDCE=\overset{s}{\underset{i=1}{\sum }}Q_{s}\times L_{s}\times \alpha_{s}\times EF_{s}\;\;\;\;\;\;\;\;\;+EF_{s2}\times E_{2}\times Q_{s2}/{Q^{\prime }}_{s2}$ $CDCE=S\times N\times EF_{0}$	Population electricity consumption, residential gas consumption, s public vehicle gasoline consumption, private vehicle gasoline consumption, heating carbon consumption
S. Zubelzu, 2015 [6]	Household-based carbon footprint calculation method	$HC\left(\frac{kgCO_{2eq}}{yr.}\right)=1.05\times [HC_{water}\;\;\;\;\;\;\;\;+HC_{wastewater}+HC_{elect.}\;\;\;\;\;\;\;\;+HC_{gas}+HC_{build.}\;\;\;\;\;\;\;\;+HC_{roadtraffic}\;\;\;\;\;\;\;\;+HC_{inhab.}]$	Drinking water consumption, wastewater management carbon, electricity, gas supply, transportation base (mandatory, non-mandatory), waste treatment in relation to urban design parameters (area, family unit) and the ability of undevelopable land to determine projected carbon emissions
Congrong, 2018 [5]	Methodology for calculating carbon emissions associated with urban sites	$CE=CR_{\mathrm{housing}}+C\_I_{\mathrm{industry}}\;\;\;\;\;\;\;\;+CI_{\mathrm{commercial}\;\mathrm{services}}\;\;\;\;\;\;\;\;+CT_{\mathrm{transportation}}\;\;\;\;\;\;\;\;+C\_r_{\mathrm{waste}}\;\;\;\;\;\;\;\;+C\_w_{\mathrm{water}\;\mathrm{resources}}$	Housing, industry, commercial services, transportation, waste, water resources
Chao Liu, 2017 [4]	Calculation of urban CO_2_ emissions	CE=BCE+TCE−GCF	Architecture, transportation, urban green space system

Carbon assessment software has improved the versatility of quantitative calculations and CO_2_ assessment methods for urban design scenarios compared to traditional formula calculations, such as the City Energy Analyst (CEA) architecture evaluation system developed by ETH Zurich. CEA integrated urban planning, thinking and energy systems technology in a comprehensive evaluation system and realized the adjustment of urban design scenarios and the calculation of urban carbon emissions to design plans and calculate energy infrastructure layouts as well as urban carbon emissions. However, most platforms are based on sensor-based monitoring of carbon emission and sink and present the data as a platform, which only applies to some cities. Trimble’s Tekla 2022, Sefaira, RIB’s iTWO 4.0 and GBSWARE’s CEEB 2023 allow for the real-time modification of building design materials and structures, carbon emission calculations and comparative calculations of multiple building design options. Systems (including control strategies), the integration of local energy resources and the intervention of regional energy systems. It also allows for urban form modifications such as new zoning, and changes in occupancy and building type, allowing for real-time carbon evaluation calculations. The Tupau carbon monitoring platform developed by Xiamen Tupau Software aims to detect carbon emissions in real time. In addition, some carbon assessment software also incorporates models of carbon interventions, such as the carbon emission measurement system of the China Institute of Planning with regional boundary selection and carbon reduction measures selection, thus performing carbon evaluation. However, most of the existing carbon assessment platforms are limited to the visualization of urban carbon emission, without considering the realization path to reach the total demand of urban carbon sink and carbon reduction and without considering the comprehensiveness of interventions and the combination scenarios of multiple approaches (Table 2).

In addition to the disadvantages of the above two approaches to urban CO_2_ assessment, more importantly, the spatial calculation scales to which the above-mentioned calculation systems are adapted are mostly city or local neighborhood-based and cannot be applied to small-scale urban design CO_2_ calculations. Furthermore, both the calculation systems and the intervention systems lack standardized spatial unit calculations and cannot accurately consider the results of local spatial carbon assessment in cities. It is not possible to accurately assess and then precisely manage, and it may also lead to underestimation or overestimation of the reliability of carbon reduction interventions. In addition, the intervention of carbon reduction studies does not consider the combination of carbon source reduction and carbon sink measures. The specific intervention measures are not directly related to the urban spatial design carriers in the intervention design study, which cannot effectively guide the urban design.

Given the above, the objectives of this study are: (1)To reconstruct a carbon-neutral assessment system at the scale of urban building clusters;(2)To realize a portfolio of interventions for urban planning with the goal of carbon neutrality;(3)To provide practical theoretical support for urban small-scale CO_2_ assessment and interventions by concretely locating the assessment results and intervention design results in urban spaces.

The innovation points of this paper are:Fine assessment of carbon emission and carbon sink in 100m*100m urban building complex space, which is convenient for planners and decision-makers to compare carbon indicators in different scales of space visually.Take more comprehensive carbon reduction measures to achieve the goal of carbon neutrality in urban building cluster area and locate the specific space, and put forward refined suggestions for the planning and implementation of locating.With the perspective of carbon neutrality, we make suggestions on industrial transformation, energy use types, ecological space layout and green building integration.

## 2. Methodology and Data Preparation

### 2.1. Data Source

Lihu Future City is located in the middle of Wuxi City, Jiangsu Province, China. The core area of Lihu Future City is about 3.7 square kilometres, located at 120°14′8″–120°15′32″ E and 31°31′6″–31°32′32″ N. Lihu Future City will carry the science and innovation exchange hub functions of the city’s science and innovation space, landscape tourism and leisure around Taihu Lake. Therefore, it is vital to achieving the goal of carbon neutrality in the future city of Lihu to construct the new town (Figure 1).

This paper collected the traffic network, land use, building-related and functional zoning data of the first round of preliminary plan of Lihu Future City. All data came from the “Wuxi Lihu Future City Master Plan”. The road network includes main roads, secondary roads, branch roads, internal access roads, etc. The land use includes residential-led composite land, public service facility-led combined land, commercial business and residential-led composite land, commercial business-led composite land, technology R&D-led composite land, parks, green areas and functional zoning. The land use includes residential-led complex land, public service facility-led complex land, commercial and residential-led complex land, commercial and commercial-led complex land, technology R&D-led complex land, park green space, square land, development reserve land and water area. Building-related data include building height data, building area data and building profile data. There are seven types of functional zoning data: central green axis, TOD complex, commercial and residential complex, waterfront commercial, science and technology-led public, science and technology-led home and public services (Figure 2 and Figure 3).

**Figure 2 ijerph-20-02420-f002:**
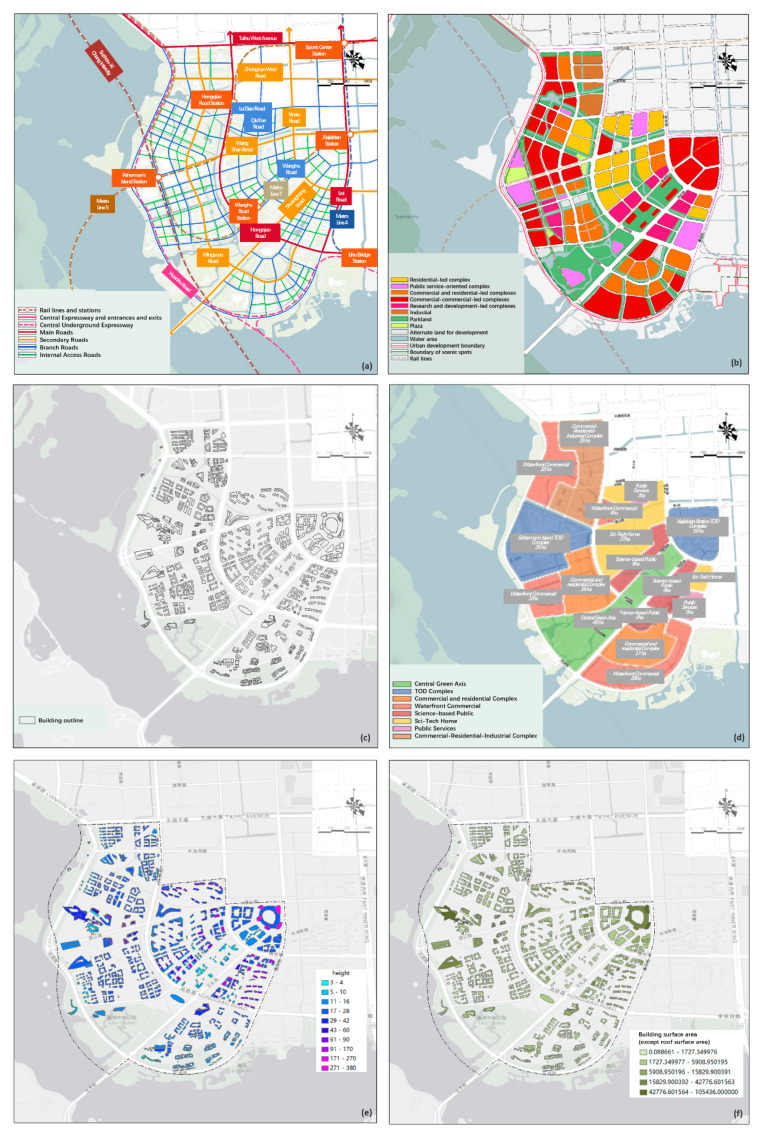
Data preparation: (**a**) road network, (**b**) land use, (**c**) architectural scheme, (**d**) functional zoning, (**e**) building height, (**f**) building surface area (except roof surface area) [24].

**Figure 3 ijerph-20-02420-f003:**
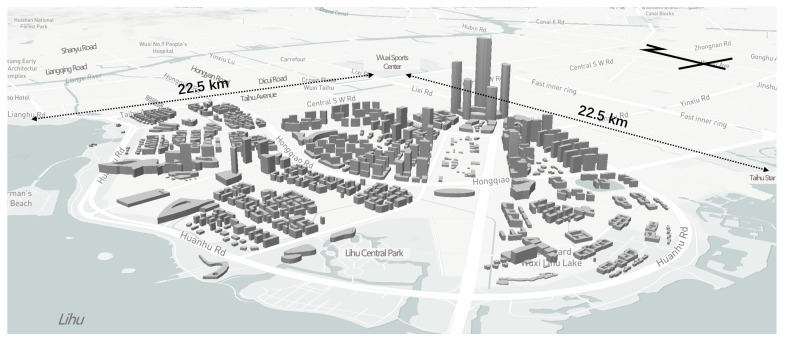
Aerial view of architectural scheme.

### 2.2. Methodology: Optimization of Carbon-Neutral Target Pathways through Design at the Scale of Urban Building Clusters

To achieve carbon neutrality at different scales of urban clusters in the design process, the calculation system of carbon assessment and interventions is reconstructed: The classification of assessment factors in the calculation system needs to be generalized to spatial units at any scale. In addition, the calculation modules and modalities can be applied to spatial elements in spatial units as well as interventions. The design of intervention measures does not only stop at strategy formulation but also needs to estimate the carbon reduction or sink potential of carbon intervention measures in a quantitative way and to consider the carbon intervention methods that can be carried out for the whole category and the whole element carrier on the basis of quantitative assessment, such as to consider different combination scenarios of both carbon reduction and carbon sink measures in a comprehensive way. The assessment results and quantitative intervention design are located in the smallest standard spatial unit so that the assessment results and intervention results can be seen visually in order to provide intuitive and effective design strategies for different urban scale planning and design.

#### 2.2.1. Reconstructing Carbon Neutral Assessment Models for Urban Building Clusters

The units and calculation methods of the traditional index evaluation system are not uniform and cannot be evaluated for all cities [25]. Building clusters are different from single building city scale or even larger scale. Building clusters are the constituent units of cities, which include all the elements of the city, but enough to effectively link with micro spaces, thus making it easy for urban planners and urban decision makers to analyze urban space at multiple scales and make detailed decisions (Table 3). Therefore, planning and design from a carbon-neutral perspective at the building community scale is one of the most effective ways to address the challenges of urban carbon reduction. Hence, it is crucial to construct a reasonable carbon-neutral evaluation model for urban building clusters [26].

The building community scale aggregates the carbon information of spatial carriers, such as buildings, building operations and traffic flows between clusters. Accordingly, this study summarized urban carbon sources and organized the methods of urban carbon emissions calculation, and divided urban carbon sources into three major categories of urban carbon emissions. The first type of urban carbon emissions is carbon emissions generated from human use, mainly from building maintenance and operation, such as carbon emissions generated from maintenance and energy consumption of the building itself, e.g., air conditioning, ventilation, heating, lighting and elevator operation. The second type of urban carbon emissions is carbon emissions generated from industrial energy consumption, mainly from the building due to commercial use. The third category is carbon emission from travel, mainly from building clusters and carbon emission from transportation service facilities such as the subway and buses between buildings and external locations. Based on the above carbon emission classification method, we established an integrated framework of carbon emission and carbon sink calculation methods to evaluate carbon emission and carbon sink of the urban design scheme and built a carbon neutral evaluation system for urban design schemes (Figure 4). At the same time, the carbon neutral results of the design scheme were calculated dynamically according to interventions of carbon reduction to realize carbon neutrality targets in the urban planning phase (Figure 5).

According to the three types of urban carbon sources mentioned above, three types of carbon emission studies were included in this study, which was building carbon emissions from human use ($CE_{H}$), carbon emissions from industrial energy consumption ($CE_{I}$) and carbon emissions from travel ($CE_{T}$). The sum of these three types of carbon emissions (CE) was used as an indicator to assess the carbon emissions of the building cluster. The basic calculation model for carbon neutral assessment can be expressed as follows:(1)CN(tyr.)=CE(tyr.)−CS(tyr.)

This study incorporated urban natural carbon sinks and carbon source reduction as the intervention measures. The spatial model was loaded with energy carbon reduction, industry carbon reduction and green space carbon sink interventions to simulate the effects of different intervention conditions in urban planning and design on the changes in carbon emission and carbon sink of the design scheme. In addition, the carbon reduction and sink measures were specifically located in four urban design elements: buildings, roads, water systems and land use.

The basic equation for the carbon neutral interventions in the model is as follows:(2)$$CE=X+\sum_{i=0}^{n}f\left(w_{i},v_{i}\right)$$
(3)$$X=CE_{H}+CE_{I}+CE_{T}$$

Here, X represents the baseline carbon emission projection, $CE_{H}$ is the carbon emission generated by anthropogenic use of buildings, $CE_{I}$ is the carbon emission generated by industrial energy consumption,$\;CE_{T}$ is the carbon emission generated by travel and $f\left(w_{i},v_{i}\right)$ represents the rules of carbon reduction or sink measures of i categories.

Carbon emissions from building operations $CE_{H}$ were calculated based on the carbon emissions per unit space of public buildings nationwide, including residential buildings, commercial buildings, cultural buildings, educational buildings, research buildings, sports buildings, medical buildings, facility buildings and industrial buildings. Some studies have shown that the average energy consumption per unit of industrial land in China is 10^4^/hm^2^ [27]. Due to the lack of average carbon emission per unit area of industrial buildings (Table 4), this study calculated carbon emission per unit area of industrial land as carbon emission per unit area of building and calculates carbon emission from industrial buildings based on the carbon emission coefficient of energy type. Carbon emissions from industrial energy consumption was calculated according to the type of energy required by the type of industry, mainly for the carbon emissions generated by the energy structure required in the production process of industrial buildings, and the expression is as follows:(4)$$CE_{H}=BE_{i}\times S_{i}$$

Here, BE represents the carbon emission coefficient per unit area of class i building, S represents the floor area of class i categories.
(5)$$CE_{T}=\frac{E_{i}}{\sum_{i}E}\times 10^{4}\times S\times EE_{i}$$

Here, $E_{i}$ represents the total energy consumption of category i, $\underset{i}{\sum }E$ represents the total energy consumption, S represents the floor area, and $EC_{i}$ represents the carbon emission coefficient per unit of energy of category i.

Travel carbon emissions $CE_{T}$ were calculated according to traffic carbon emissions of different road types, including collector road, arterial road, major road, highway and expressway, and the disparity in carbon emissions by road types is shown [28] (Table 5). Changzhou is a typical case of a developed city in eastern China, with a similar level of development to Wuxi, so this study cited Changzhou city related transport carbon emissions for the study (as shown in the table), and the expression is as follow:(6)$$CE_{H}=TE_{i}\times T_{i}$$

Here, TE represents the carbon emission coefficient per unit length of class i road and T represents the length of class i road.

In terms of carbon sink (CS), referring to the IPCC and existing relevant research [29,30,31], the carbon sink coefficients of different land-use types were determined (Table 6) [32]. The vegetation layer and the soil layer of green roofs can capture carbon dioxide from the air through photosynthesis, so green roofs can be one of the effective paths to reduce urban carbon dioxide emissions [33]. Vertical greening system is a fixed plant structure on the facade of a building [34]. The model focused on the city’s carbon sink, including the carbon absorption and conversion of green space systems such as cropland, woodland, garden, grassland, wetland, rooftop greening and façade greening. The expressions are as follows:(7)$$CS=\frac{1}{1000}\sum^{\;}A_{i}\cdot {\mathrm{EF}}_{i}$$

Here, *CS* represents carbon sequestration by greenland carbon sink, tCO_2_/a, A_i_ represents the area of the i type of greenland, m^2^, ${\mathrm{EF}}_{i}$ represents the carbon sequestration coefficient of the i type of greenland, kg CO_2_/m^2^.

**Table 6 ijerph-20-02420-t006:** Carbon emission factors for different types of green spaces.

Greenland Use Type	Carbon Absorption Coefficient (kg CO_2_/m^2^)	References
Forestland (Zhejiang, China)	9.7	Yin, Gong et al. (2022) [35]
Grassland	0.021	Zhang et al. (2018) [29]
Garden plots	0.1847	Zhang et al. (2015) [29]
Roof garden	1.35	Cascone, Catania et al. (2018) [36] [37]
Three-dimensional greening	4.042	Marchi, Pulselli et al. (2015) [38] Shafique, Xue et al. (2020) [37]

#### 2.2.2. Selection of Carbon Reduction and Sink Measures

Corresponding to the three types of urban carbon sources, the choice of carbon reduction technologies should revolve around the process of carbon element flow from source emission to end-use absorption. The measures focused on four clusters of carbon reduction technologies in building clusters, including building energy reduction technologies, industrial structure upgrading, infrastructure carbon reduction technologies and greenland carbon sink system enhancement technologies. This study focused on specific leading-edge technology types widely used in these four building complexes. We discussed the measurement of their carbon reduction potential and related the leading distribution vectors of specific technology types to the design elements of urban building complexes (Figure 6). Finally, we tailored the selection of effective carbon reduction interventions for different design solutions.

In terms of building energy carbon reduction technologies, the primary energy carbon reduction technologies suitable for small-scale building clusters included solar panels, geothermal heat pumps (GHP) and biodigesters (biomass fuel) (Table 7). Solar panels and heat pump technologies use renewable energy sources such as solar and geothermal energy to reduce carbon emissions associated with traditional coal-fired power generation. Based on the choice of energy supply types and their average CO_2_ potential, the urban building stock scale interventions and plans to achieve carbon neutrality targets are developed. Similarly, in terms of infrastructure carbon reduction technologies, energy savings in lighting facilities such as LED lighting, which can increase the utilization of renewable energy supplies, should be considered. China’s power energy structure is similar to that of India, mainly thermal power generation and coal power generation. Therefore, based on The installation of 1 million LED streetlights has resulted in annual energy savings of 6.71 billion kWh and avoided 1119.40 MW of peak energy demand, resulting in a reduction of 4.63 million tons of CO_2_ emissions per year of the Indian Plan [39]. Thus, one sets LED streetlights reduces carbon emissions by an average of 9.26 tons of CO_2_ emissions per year.

In terms of carbon technology in the construction industry structure, carbon emission intensity (CI) refers to carbon emissions per unit GDP, which can reflect the actual economic development and CO_2_ emissions of a country (region) more scientifically than carbon emissions [40]. Industrial restructuring is one of the critical means to achieve economic transformation and reduce carbon emissions [41]. From the general rule of industrial structure evolution, the shift of industrial structure center of gravity from primary to tertiary industries is often accompanied by the gradual replacement of traditional industries by low-pollution, low-energy-consuming and high-value-added new industries, which not only improves factor production efficiency but also reduces energy consumption intensity, thus positively influencing the reduction of carbon emission [42,43,44]. This study prioritized the selection or upgrading of industry types in the production category by calculating the quantitative values of two indicators. Industrial structure optimization had a significant negative impact on carbon emission intensity. After controlling for the relevant variables, every 1% increase in industrial structure optimization reduced the carbon emission intensity by 1.117% [45]. Therefore, it is recommended that Chinese provinces continue accelerating the pace of industrial transformation and upgrading, promote the industrial structure in the direction of advanced development and encourage the transformation and upgrading of traditional industries. In this way, they can improve industrial efficiency and strive to achieve low energy consumption and high output. It is crucial to accelerate the formation of knowledge and technology-intensive industries such as strategic emerging and high-tech industries [45].

In terms of greenland carbon sink system, the key to carbon reduction was to increase the area of green space scale and consider the carbon sink efficiency of plant types. For example, one hectare (2.5 acres) of forest can absorb 1.5 to 30 metric tons (1.6 to 33 tons) of carbon dioxide per year, but the carbon sink of the same area of space grass is one per cent or even one thousand of that of trees. Therefore, the construction of terrestrial green space ecosystems should give priority to the planting technology of trees or unique plants that can efficiently carry out carbon sink or the optimization technology of the original vegetation. In addition, the choice of carbon reduction technology for building clusters can consider three-dimensional greening and rooftop greening technology. The increase in three-dimensional greening and rooftop greening in densely built areas is relatively limited in terms of the increase in total carbon sink. However, a rough estimate of the potential area range of greenery on the facade (excluding windows) is twice the ground footprint of buildings within the city. In comparison with green roofs, an even more tremendous potential to realize the environmental benefits of vegetated building surfaces with facade greening. VGS offer multiple benefits as innovative components of urban design and increased greenery in built environments lacking green areas [46]. The carbon reduction potential of carbon sink enhancement technologies was measured by calculating the new plant types and carbon sink efficiency of the increased greenery and rooftop greenery areas.

#### 2.2.3. Carbon Neutral Assessment and Intervention Results at the Scale of Urban Building Cluster

The assessment model should divide the urban design scheme into a standardized calculation grid to accurately assess the city carbon emission and carbon sink values at the scale of urban building clusters. The Inverse distance weighted method (IDW) can predict the overall spatial distribution and has higher prediction accuracy with limited sample points than other difference methods [47]. Based on the arcmap platform, the Inverse distance weighted method is used to assign the carbon emission and carbon sink values to a 100m*100m grid. The assessment scope can be freely defined, which can help to assess the carbon emissions and carbon sinks of different design scales, such as single buildings and building clusters. In addition, the layout of interventions also falls to specific buildings, roads and ground, and the carbon emissions and sinks of the grid change with the type of interventions and the number of interventions stacked after the interventions are set.

## 3. Results and Discussion

### 3.1. Spatial Distribution of Carbon Emissions

Based on the existing ecological foundation layout scheme, the total carbon sink of Lihu Future City was about 116,000 tons of CO_2_, the total carbon reduction was about a mere 0.1 million tons of CO_2_, and there was a gap of about 115,000 tons of CO_2_ from the carbon-neutral target. It was mainly distributed in the Xiajiabian Station TOD Complex, which accounted for 26% of the total carbon emission, and the Fisherman’s Island TOD Complex, which accounted for 25% of the total carbon emission, followed by industrial and Charming Creative Neighborhood, each accounting for 6% and 4% of total carbon emissions. The central carbon sink areas were concentrated on the waterfront shoreline accounting for 48% of the whole carbon sink, and the central garden axis accounted for 27% of the whole carbon sink (Figure 7, Figure 8 and Figure 9).

The top 5 most urgent carbon reduction 100 × 100 m spaces of Lihu Future City were distributed in the Xiajiabian Station TOD Complex area and the Fisherman’s Island TOD Complex. Within the area of the Xiajiabian Station TOD Complex, the space most urgently in need of carbon reduction is space 1, which has a carbon emission of approximately 5000 tons, accounting for about 16% (Figure 10a). Within the area of the Fisherman’s Island TOD Complex, the space most urgently in need of carbon reduction is space 1, which has a carbon emission of roughly 3000 tons, accounting for about 10% (Figure 10b).

### 3.2. Realizing the Spatial Location of Carbon-Neutral Measures

In the layout of interventions in Lihu Future City, based on the base site attributes, current conditions of building density, and a large carbon neutral gap, nine types of carbon reduction measures were considered in this study, namely dense forest planting, three-dimensional greening technology, green roof system, solar panel system, LED street lights, biodigester, conversion of intensive commercial areas into research-led industries and upgrading industrial enterprises into research pilot industries (Figure 11 and Figure 12).

The carbon sink measures were mainly distributed in the Xiajiabian Station TOD complex, the charming creative block, the lakeside green corridor, the Fisherman’s Island TOD complex, the three-dimensional composite landscape green axis and other open space areas to increase partnership dense forest planting. The carbon sink potential reached about 16,000 tons, accounting for about 14% of total carbon reduction. The increase in three-dimensional greening measures in the façade space of buildings across the region and the growth of green roof technology, comprehensive carbon sink and carbon reduction potential in the roof space of buildings across the region reached 15,500 tons, accounting for about 14% of the whole carbon reduction. The solar panel power supply technology, ground-source heat pump heating technology and bio-digester heating technology for nearly the whole area of the building, and the arrangement of LED street light sets on Wang Shan Road, Yinxiu Road, Shuanghong Road, Wang Hu Road, Huanhu Road and Hongqiao Road, reduced carbon by a total of 54,000 tons, accounting for 47% of the whole carbon reduction, and were the most effective types of carbon reduction measures. By upgrading industrial to scientific experimental industries and converting waterfront commercial dense areas to scientific experimental industries, a total of 3000 tons of carbon was reduced, accounting for 25% of the whole carbon reduction. Thus, based on the layout of the nine types of carbon reduction measures, the design scheme of Lihu Future City could reach the carbon-neutral target (Table 8, Figure 13).

Studies have been conducted to investigate the relationship between the national level, city level and individual elements of the city, such as separate buildings and transportation, and carbon emissions. However, more research needs to be carried out to calculate and comprehensively lay out carbon reduction and sink measures in advance at the urban planning stage with carbon neutrality targets and to locate specific spaces. However, it is crucial to lay out carbon source reduction and carbon sink measures in advance in the planning and design stage. For example, as we can see in the above project application, if we need to achieve the city’s carbon neutrality target, more than 50% of the building facades need to install the three-dimensional greening system, and more than 70% of the building roofs need to lay out green roofs (Table 8, Figure 14).

In the planning and design stage, carbon indicators are quantitatively calculated for each element of the city. The division of fixed calculation units allows designers to consider carbon metrics at different scales in the design process and thus obtain a specific carbon-carbon neutral gap. After receiving the carbon-neutral opening, it is easier for designers to check the interaction between carbon targets and building design, industrial layout, transportation planning, etc., through iterative design solutions. This iterative process allows urban planners to deploy carbon reduction and sink measures in detail at the design stage. By differing from existing carbon prospective studies, our study locates specific design elements and flexible small-scale units of carbon emissions in urban design to support urban planners in the design process to achieve carbon neutrality goals. For example, in the intervention mentioned above layout of carbon reduction measures in Lihu Future City, the layout of clean energy carbon reduction can lead to a sharp decrease in urban carbon emissions (Figure 13). Still, conversely dense commercial-dominated areas, as well as industrial areas, have a positive impact on carbon emissions (Figure 14).

This paper only considered one scenario in which the new city achieves the carbon neutrality goal. It is worth exploring whether the new town has a more significant potential for carbon reduction. In our future research, we will investigate whether there is potential for larger-scale carbon reduction targets based on the city’s characteristics and current situations. In addition, the choice of carbon reduction interventions should also consider their cost. For example, the average total costs of installation, operation and maintenance for green roof and 3D green system installations were 103.28 €/m^2^/year and 195.57 €/m^2^/year, respectively [48], while the average total cost of the initial system cost and the annual operation of ground source heat pumps was 15.73 €/m^2^/year [49]. Therefore, if cost alone is used as the evaluation criterion for selection preference, priority should be given to installing ground-source heat pump systems. However, the selection priority of interventions should be based on their carbon reduction efficiency and the degree of enforceability in the sample area. Accordingly, we will consider combining interventions based on cost setting and carbon reduction targets in a site-specific manner to achieve urban sustainable development goals more effectively. Furthermore, this paper’s carbon emission and carbon sink coefficients were average values. The coefficient calculation will be further studied and optimized in the following research plan by considering urban clusters’ geographical characteristics and development.4. Conclusions

The dimensions of carbon assessment in existing studies are primarily urban scale and urban macro-level elements, without careful consideration of carbon emission values generated by human use, industrial types, transportation, etc., in building complexes. Interventions are mostly building or energy-specific studies, lacking comprehensiveness and the assessment of the benefits of interventions needs to consider the effects of urban building clusters. The carbon assessment results and interventions cannot be localized in urban spaces on a small scale and cannot achieve multi-scale spatial carbon neutrality assessment. Based on the above gaps and challenges, we aimed to carbon assess multi-scale urban spaces and to lay out carbon reduction interventions in advance at the urban design stage to achieve the goal of carbon neutrality in urban building clusters.

To achieve our research objectives, in this paper, we proposed three optimization methods and used Lihu Future City as our practice case. Firstly, the assessment and intervention system was reconstructed to be able to be used to assess multi-scale spatial carbon indicators; secondly, on this basis, carbon reduction measures suitable for multi-scale space were summarized and divided into four categories: building energy saving and emission reduction technologies, building industrial structure carbon reduction methods, infrastructure carbon reduction technologies and green space carbon sink systems for building clusters, and their specific categories and calculation methods were detailed, respectively, and finally, the specific intervention measures were matched with urban design elements match, and established the “technology-spatial” intervention design model; thirdly, established the unit calculation method, and placed the assessment results and intervention results specifically in the 100 × 100 m urban space unit. Finally, we took the Lihu Future City as a practical example for explaining how to apply the three optimal paths to achieve the carbon neutrality goals of different scales of urban design. For urban planning and design, it is of great significance to dynamically assess the carbon emission and carbon sink of urban schemes during the design process. It is also crucial to make comprehensive carbon reduction and carbon sink interventions for urban design elements based on the assessment data results and to obtain the evolution trend of carbon emission and carbon sink data in real-time for the carbon neutral target.

This paper has three contributions: We reconstructed the computational model and the quantitative intervention system in order to apply to the quantification of carbon indicators in multi-scale urban design.We more comprehensively considered carbon reduction measures that can be applied to urban design elements.We specifically located the interventions in the 100 × 100 m urban space and design elements, thus proposing the most direct and effective recommendations for carbon neutrality in urban design schemes.

Due to the lack of field validation of the CO_2_ detection data and potential quantitative data for the intervention, it is not possible to assess the accuracy of the relevant data and coefficients presented. In future studies, we will calibrate the parameters involved in the method with measured data to obtain more accurate results.

## Figures and Tables

**Figure 1 ijerph-20-02420-f001:**
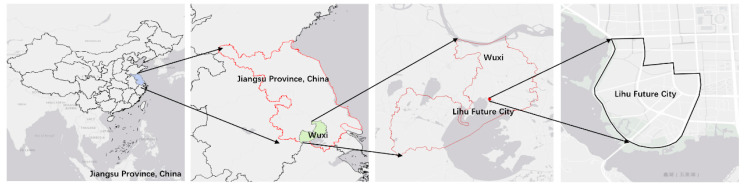
Lihu Future City Location Map.

**Figure 4 ijerph-20-02420-f004:**
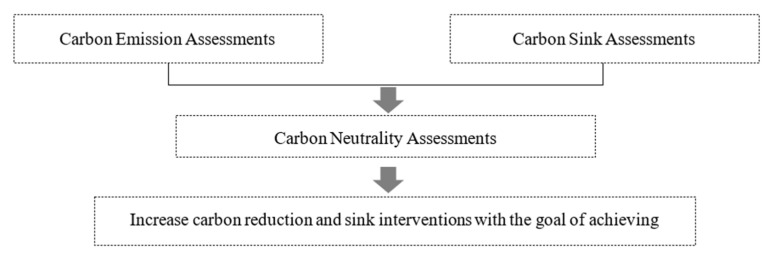
Carbon Evaluation and Intervention Theory Mode.

**Figure 5 ijerph-20-02420-f005:**
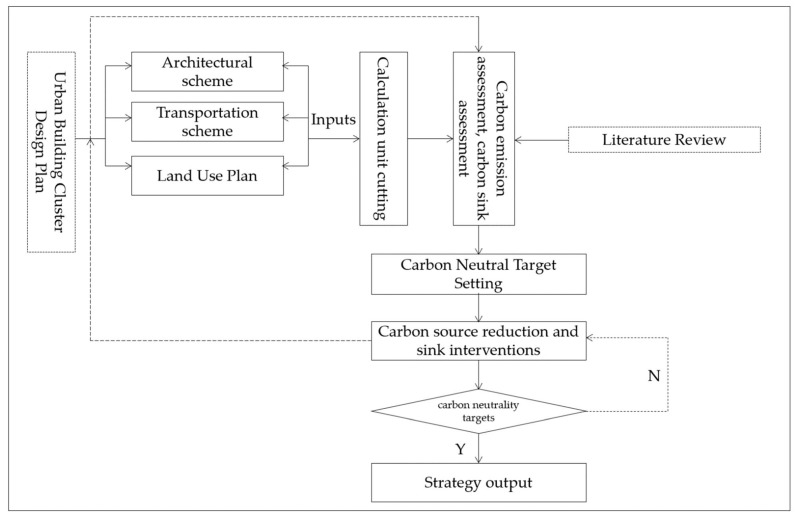
Carbon evaluation and intervention model design.

**Figure 6 ijerph-20-02420-f006:**
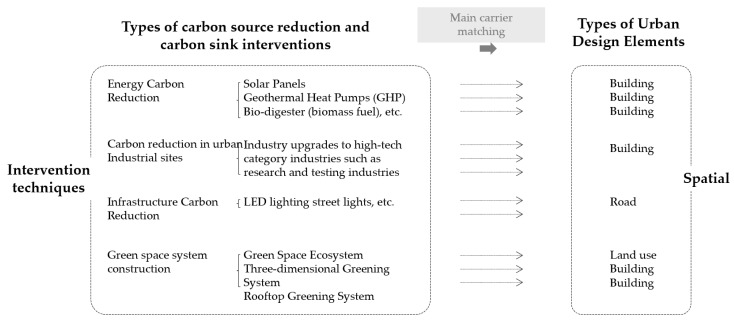
Matching carbon reduction and carbon sink approaches with urban design elements.

**Figure 7 ijerph-20-02420-f007:**
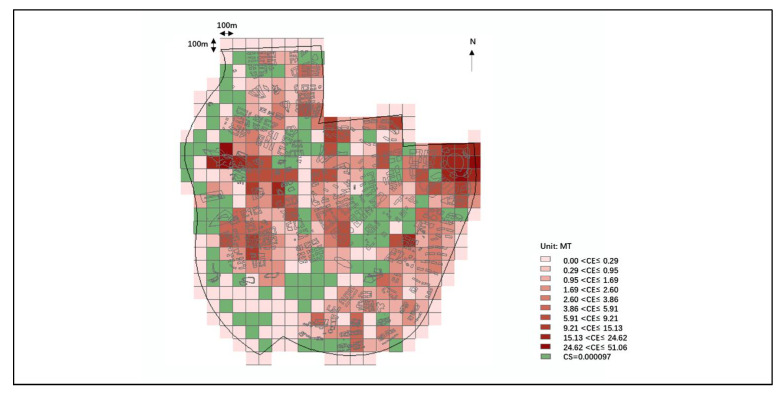
Carbon emission and carbon sink distribution of Lihu Future City design scheme in 100 × 100 m unit.

**Figure 8 ijerph-20-02420-f008:**
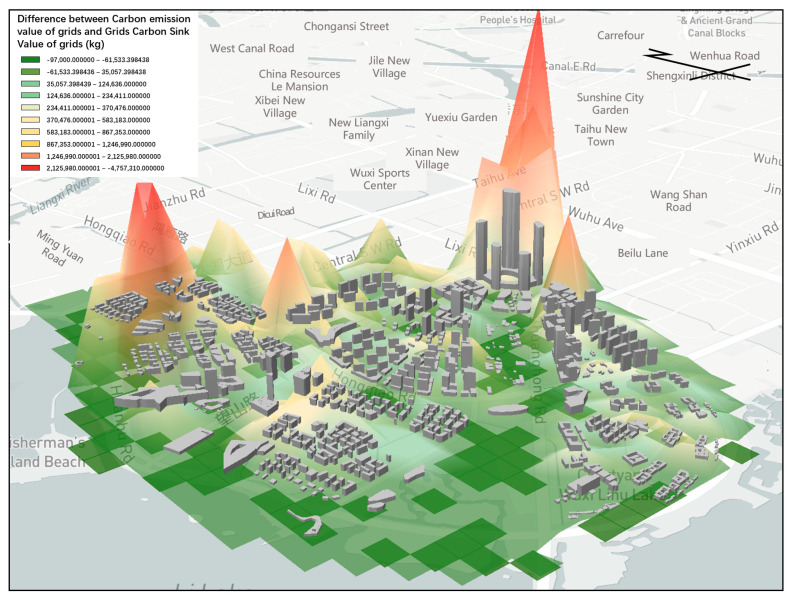
Carbon emission and carbon sink distribution of Lihu Future City design scheme (Visualization based on mapbox platform).

**Figure 9 ijerph-20-02420-f009:**
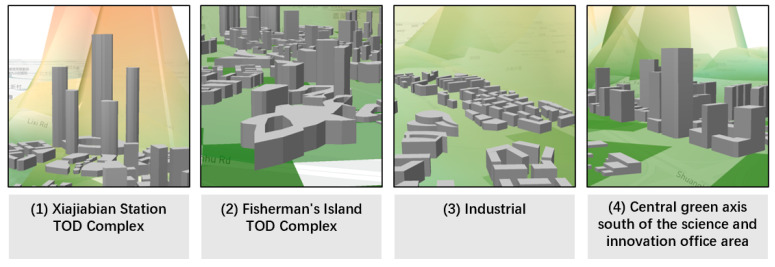
Areas with high carbon emissions (Visualization based on mapbox platform).

**Figure 10 ijerph-20-02420-f010:**
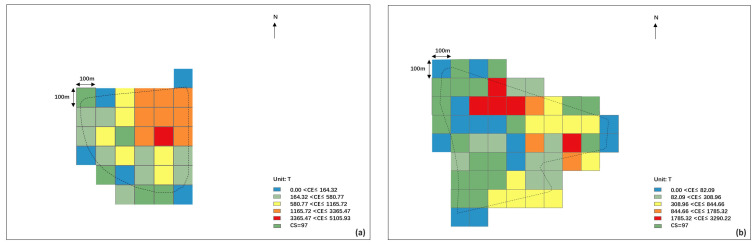
(**a**) Carbon emission and carbon sink distribution of the Xiajiabian Station TOD Complex in 100 × 100 m unit, (**b**) Carbon emission and carbon sink distribution of the Fisherman’s Island TOD Complex in 100 × 100 m unit.

**Figure 11 ijerph-20-02420-f011:**
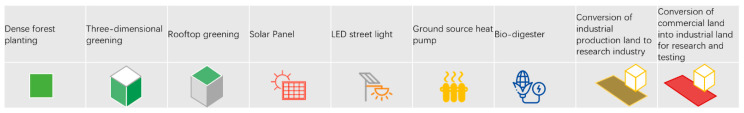
Diagram of different types of interventions.

**Figure 12 ijerph-20-02420-f012:**
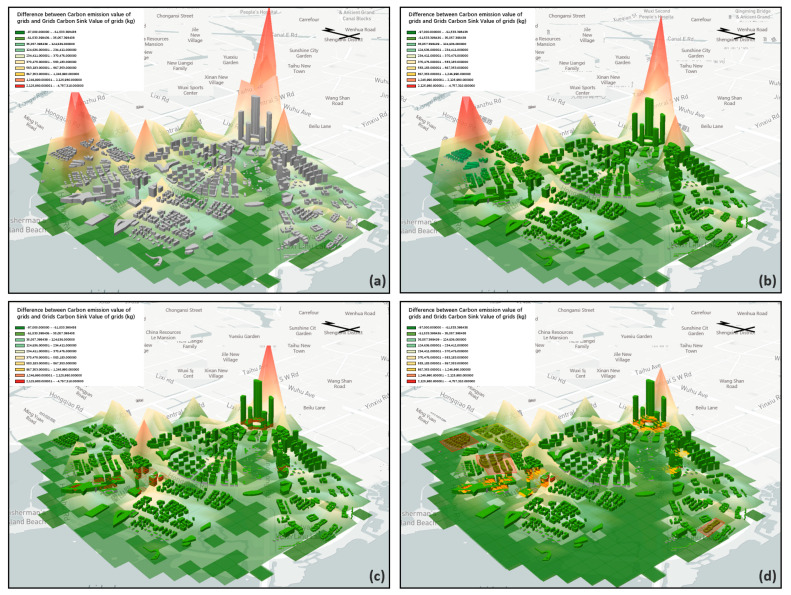
Carbon emission and carbon sink distribution of Lihu Future City design scheme: (**a**) Initial Carbon Sink and Carbon Emission, (**b**) Diagram of green roof interventions and three-dimensional greening intervention, (**c**) Diagram of the overlay of green roof interventions, three-dimensional green interventions and solar panel interventions, (**d**) Overlay diagram of green roof interventions, three-dimensional green interventions, solar panel interventions, ground source heat pump interventions and transformation of industrial industries and waterfront commercial areas into research and experimental industrial areas (Visualization based on mapbox platform).

**Figure 13 ijerph-20-02420-f013:**
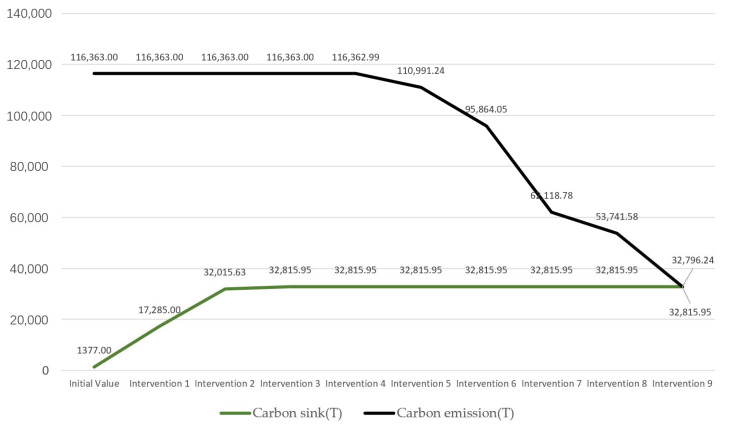
Change in programmatic carbon emission and sink values in the process of carbon neutrality achieved by stacking interventions.

**Figure 14 ijerph-20-02420-f014:**
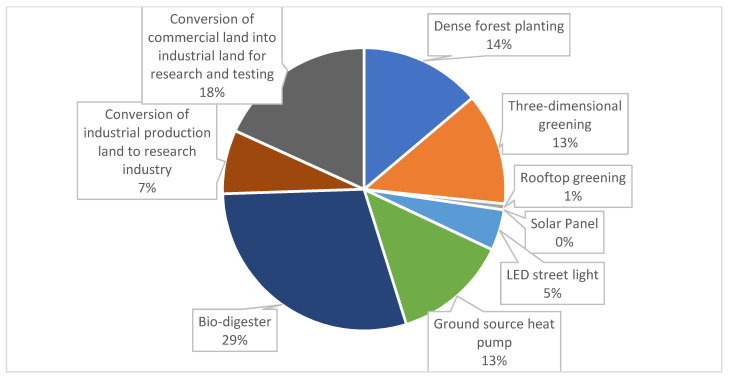
Share of carbon reduction potential for each type of intervention.

**Table 2 ijerph-20-02420-t002:** Comparison of the functions of existing carbon assessment platforms.

	Intelligent Clusters Lifecycle	Take 2022 Sefaira Itwo4.0CEEB2023	City Energy Analyst	Cooling Singapore	Carbon Emission Measurement System of China Institute of Planning	Tupou Carbon Monitoring Platform
Carbon Emission Assessment	Yes	Yes	Yes	Yes	Yes	Yes
Carbon Sink Assessment	Yes	Yes	Yes	Yes	No	No
Scope of intervention dynamic assessment	No	No	Yes	No	No	No
Increased manual interventions for carbon reduction and real-time assessment	Yes	Yes	No	No	Yes	No
Increased manual interventions for carbon sinks and real-time assessments	Yes	Yes	No	No	No	No
Carbon reduction weights comprehensiveness	Low	Low	Low	Low	Medium	Low
Sinks carbon weights comprehensiveness	Low	Low	Low	Low	Medium	Low
Data Analysis Visualization	High	Medium	Medium	High	Low	Low
Convenience of use process	High	Low	Medium	High	Medium	Low
Platform Shareability	Low	Medium	High	High	Low	Low

**Table 3 ijerph-20-02420-t003:** Urban Building Cluster Research Scale.

Urban Building Cluster Research Scope	Scale	Carbon Neutral Units
National, city, district and county-wide	——	MT
Urban design (small scale)	>500,000 m^2^	MT
Partial Area	>40,000 m^2^, <=500,000 m^2^	10 KT
Single Building	<=40,000 m^2^	T

**Table 4 ijerph-20-02420-t004:** Carbon emissions per unit of floor space.

Type of Land Use	Building Type	Building Specific Categories	Carbon Emission (kg CO_2_/m^2^)
Residential, Commercial and Residential	Residential buildings	Residential buildings in cities and towns	90.79
Administration	Office buildings	Government office buildings	65.12
Medical Land	Medical buildings	Hospitals	76.26
Sports	Sports	Sports Complexes	89.66
Commercial	Commercial buildings	Hotels and restaurants	109.11
Commercial, Commercial and Residential	Commercial buildings	Supermarkets	122.99
Commercial	Commercial buildings	Shopping mall building	179.15
Cultural, Education and Research	Cultural buildings, Research buildings	High school building	40.59
Covering all land types	Office buildings	General office buildings	64.08

**Table 5 ijerph-20-02420-t005:** Carbon emission factor per unit of road.

Road Type	Carbon Emission (t)	Carbon Emission (kg per m)
Collector road	1099.67	1.53
Arterial road	1416.39	1.96
Major road	167.73	0.40
Highway	79.65	0.58
Expressway	84.39	0.41

**Table 7 ijerph-20-02420-t007:** Summary of quantitative calculations of the relationship between renewable energy and carbon reduction potential.

Type of Energy Supply	Average Value of CO_2_ Reduction Potential
biodigesters (biomass fuel)	−29%
solar panels on 50% of the roof area to generate electricity	−4%
geothermal heat pumps (GHP)	−13%

**Table 8 ijerph-20-02420-t008:** Specific information on carbon source reduction or carbon sink interventions.

	Types of Interventions	Specific Distribution Location	Quantity	Unit	Carbon Sink/Carbon Reduction (T)	Carbon Sink/Carbon Reduction (T)	Carbon Reduction Pathways
Intervention1	Dense forest planting	Xiajiabian Station TOD Complex	40,000	m^2^	388	15,908	Carbon Sink
		Charming Creative Neighborhood	700,000	m^2^	6790		
		Lakeside Green Corridor	200,000	m^2^	1940		
		Fisherman’s Island TOD Complex	200,000	m^2^	1940		
		Three-dimensional composite landscape green axis	500,000	m^2^	4850		
Intervention2	Three-dimensional greening	Whole area building façade	3,644,391.82	m^2^	14,730.63174	15,530.95	Carbon Sink, Carbon Source Carbon Reduction
Intervention3	Rooftop greening	Whole building roof	592,830.21	m^2^	800.3207835		
Intervention4	Solar Panel	Whole building roof	592,830.21	m^2^	0.008384313	54,244.22	Carbon Source Carbon Reduction
Intervention5	LED street light	Wangshan Road	50	set	463		
		Yinxiu Road	30	set	277.8		
		Shuanghong Road	50	set	463		
		Wanghu Road	100	set	926.1		
		Huanhu Road	200	set	1852.4		
		Hongqiao Road	150	set	1389.45		
Intervention6	Ground source heat pump	Whole Area Architecture	13%	——	15,127.19		
Intervention7	Bio-digester	Whole Area Architecture	29%	——	33,745.27		
Intervention8	Conversion of industrial production land to research industry	Industrial	117%	——	8377.2	29,322.54	
Intervention9	Conversion of commercial land into industrial land for research and testing	Waterfront Commercial Area	18%	——	20,945.34		

## Data Availability

No new data were created or analyzed in this study. Data sharing is not applicable to this article.

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
