# Peer review of "Achieving Carbon Neutrality through Urban Planning and Design"

_ijerph, 2023, doi:10.3390/ijerph20032420_

Round 1

Reviewer 1 Report

really interesting topic, returned graphically with interesting images

insert image's source, also when is autor's elaboration

some references are in written in chinese characters

is it possible to add some elaboration or consideration in the final conclusion about the necessary costs related to the 4 different scenarios of figure 11? maybe it's also interesting to define a cost benefit balance

Author Response

Thank you for your professional insight into this paper and for your comments on revisions.

Based on your suggestions, we summarized the following 3 modifications:

  1. insert image's source, also when is autor's elaboration

Response1:

In response to this suggestion, we have rechecked the articles and all image's source. The pictures in the article were made by the author of this article.

  1. some references are in written in chinese characters

Response2:

According to your suggestion, lines 570 and 590 of the thesis were modified as follows:

“5. Cong R., Research on land use change and its carbon emission effect in Jinan City. 2018, Shandong University of Finance and Economics.

15. Wang Yifan, Li Xue, and Yuan.J. Building energy efficiency, housing form and carbon emission correlation study--Jintang County as an example. 2021.”

  1. is it possible to add some elaboration or consideration in the final conclusion about the necessary costs related to the 4 different scenarios of figure 11? maybe it's also interesting to define a cost benefit balance

Response3:

We appreciate it very much for this good suggestion, lines 499 to 509 of the thesis were modified as follows:

In addition, the choice of carbon reduction interventions should also consider their cost. For example, the average total costs of installation, operation and maintenance for green roof and 3D green system installations were 103.28 €/m2/year and 195.57 €/m2/year respectively [48], while the average total cost of the initial system cost and the annual operation of ground source heat pumps was 15.73 €/m2/year [49]. Therefore, if cost alone is used as the evaluation criterion for selection preference, priority should be given to installing ground-source heat pump systems. However, the selection priority of interventions should be based on their carbon reduction efficiency and the degree of enforceability in the sample area. Accordingly, we will consider to combine interventions based on cost setting and carbon reduction targets in a site-specific manner to achieve urban sustainable development goals more effectively. 

48. Manso, M., et al., Green roof and green wall benefits and costs: A review of the quantitative evidence. 2021. 135: p. 110111.

49. Man, Y. and H. Yang. Study on hybrid ground coupled heat pump systems for cooling dominated buildings. in Liaoning (Dalian)-Hong Kong Joint Symposium. 2010."

Reviewer 2 Report

Some good visuals in the report, but written tables' wording needs some edits. Several sentences and many errors made the report difficult to read. The abstract states what was done but is less explicit about the key findings. State them more clearly to hook the reader and to provide the report's import. Consider elaborating more on what cannot be done without your building cluster method. Describe interventions that would be done at a cluster level further to increase your import. The examples selected to demonstrate the formula seem to go without saying. An industrial complex is going to have higher emissions than a park. While a cluster can see the net effect, a citywide or regional cluster could do it too. Moreover, carbon neutrality is one thing, but new cities are likely going to need to offset more carbon, not just be neutral. So, the paper is not ambitious enough in this practical sense. It seems to me that the most significant revisions that are necessary are to writing and to framing. Help the reader understand better the novelty and import of your formula. Remember to cite all claims.

Author Response

Thank you for your professional insight into this paper. Based on your comments, we have revised the paper. The details of the revisions are in the uploaded document.

Reviewer 3 Report

Good research! However, the references should be translated into English before acceptance.

Author Response

Thank you for your professional insight into this paper. Based on your suggestions, lines 570 and 590 of the thesis were modified as follows:

5. Cong R., Research on land use change and its carbon emission effect in Jinan City. 2018, Shandong University of Finance and Economics.

15. Wang Yifan, Li Xue, and Yuan.J. Building energy efficiency, housing form and carbon emission correlation study--Jintang County as an example. 2021.